# Past, Present, and Future of Genome Modification in *Escherichia coli*

**DOI:** 10.3390/microorganisms10091835

**Published:** 2022-09-14

**Authors:** Hirotada Mori, Masakazu Kataoka, Xi Yang

**Affiliations:** 1Innovation Laboratory of Systems Microbiology and Synthetic Biology, Institute of Animal Sciences, Guangdong Academy of Agricultural Sciences, Guangzhou 510640, China; 2Department of Environmental Science and Technology, Faculty of Engineering, Shinshu University, Nagano 390-8621, Japan

**Keywords:** *Escherichia coli* K-12, mutation, homologous recombination, HR, site-specific recombination, genome modification, λ Red, P1 transduction, recombineering

## Abstract

*Escherichia coli* K-12 is one of the most well-studied species of bacteria. This species, however, is much more difficult to modify by homologous recombination (HR) than other model microorganisms. Research on HR in *E. coli* has led to a better understanding of the molecular mechanisms of HR, resulting in technical improvements and rapid progress in genome research, and allowing whole-genome mutagenesis and large-scale genome modifications. Developments using λ Red (*exo*, *bet*, and *gam*) and CRISPR-Cas have made *E. coli* as amenable to genome modification as other model microorganisms, such as *Saccharomyces cerevisiae* and *Bacillus subtilis*. This review describes the history of recombination research in *E. coli*, as well as improvements in techniques for genome modification by HR. This review also describes the results of large-scale genome modification of *E. coli* using these technologies, including DNA synthesis and assembly. In addition, this article reviews recent advances in genome modification, considers future directions, and describes problems associated with the creation of cells by design.

## 1. Introduction

The discoveries of bacterial conjugation [1] and of generalized transduction [2] have enabled genetic research in *Escherichia coli* K-12. Subsequent genetic investigations of *E. coli* K-12 and its bacteriophages have increased the knowledge of gene structure and function, and have led to the emergence of molecular biology. Although DNA transfer by transformation had been previously described [3] and was shown to occur naturally in *Pneumococcus* [3], *Haemophilus* [4], and *Bacillus subtilis* [5], *E. coli* proved to be recalcitrant. Treatment with CalCl_2_ allowed the transformation (transfection) of *E. coli* with bacteriophage [6] and plasmid DNA [7], but not transformation by (linear) chromosomal DNA. Based on a hypothesis that an endogenous exonuclease in *E. coli* degrades linear DNA, *E. coli recBCD* mutants, which lack the RecBCD exonuclease, were found to be transformable by chromosomal DNA, provided the strain carried a cryptic prophage encoding the SbcBC(D) recombinase [8]. Expression of the λ Red recombinase (*exo*, *bet*, and *gam*) was shown to increase the efficiency of homologous recombination (HR) with linear DNA [9], leading to the use of λ Red recombinase in highly efficient systems for direct modification of chromosomal genes via HR [10,11].

Clustered Regularly Interspaced Short Palindromic Repeat (CRISPR)-Cas systems participate in acquired immunity in archaea and bacteria [12]. Although these unusual repetitive DNA sequences were first described in 1987 [13], molecular understanding of their function was first determined 25 years later [14], leading to dramatic advances in the technology of genome modification [15].

Genome research has become increasingly important in the 21st century. Technological innovations in the 1990s increased the efficiency and reduced the costs of genome modification and analysis, such as DNA sequencing and DNA synthesis. Construction of a minimal genome enabled improvements in the ability to synthesize antibiotics and produce other valuable materials.

To date, large-scale deletions of genes other than those that are clearly unnecessary, such as prophage, transposon regions, and insertion sequences, have been unsuccessful [16]. Knowledge of the principles of genome construction is still incomplete, even in model organisms such as *E. coli*. Although a fully chemical synthetic bacterial genome has been constructed in *Mycoplasma* [17,18], it was not completed by design. The development of whole-cell metabolic models has been steadily progressing [19,20,21,22,23], and these models have become platforms for genome design. These models, even those in *E. coli*, contain large numbers of genes with unknown or incomplete functions [24]. Metabolic and whole-cell models have been constructed for *Mycoplasma* [25] and are progressing steadily for *E. coli* [26], although additional research is required to create an *E. coli* model cell for genome design. This review summarizes the historical background of technological improvements, shows examples of past and ongoing research, and considers the current status and future of this research.

## 2. Historical Perspective of *E. coli* as a Biological Research Tool

### 2.1. Before the Molecular Biology Era

The discovery of conjugation in *E. coli* K-12 [1], which was thought to be sexless and to grow monogamously, and of transduction [2], showed that genetic traits could be transferred between bacteria, leading to the use of *E. coli* K-12 as a model cell. Studies of the molecular mechanism of conjugation showed that it required a fertility factor and that DNA is transferred by the Type IV secretion system [27], the detailed molecular structure of which was visualized by cryo-electron microscopy [28]. Results showing that genetic transformation requires extracellular DNA, not protein or RNA [3], provided proof that genes are composed of DNA 28. Moreover, bacterial species such as *Pneumococcus* [3], *Haemophilus influenzae* [4], and *Bacillus subtilis* [5] were shown to have the ability to take up extracellular DNA.

The discovery of conjugation and transduction in *E. coli* made genetics possible. Table 1 shows genes associated with recombination in *E. coli*, including genes required for HR, site-specific recombination, and transposition. Table 2 summarizes methods used in *E. coli* genome-scale studies. This review focuses on the use of HR for genome modification. HR methods based on λ Red have been expanded for genome-scale functional analyses of *E. coli* and its phages [29,30]. Selected references are in Table 2, with additional references cited in a previous review [31].

### 2.2. Genetic and Genomic Engineering in the Molecular Biology Era

Following studies showing that *recBCD* mutations suppress linear DNA degradation and *sbcA* mutations activate the *recET* pathway, HR was developed for cloning PCR products [42]. This method was modified during the development of an in vivo cloning (iVEC) protocol, which requires *xthA* (exonuclease III) and is independent of RecA and RecET [43]. Table 1 lists genes related to HR in *E. coli*. Mutations improving HR have been identified in the *recB*, *recC*, *recD*, *sbcA*, *sbcB*, *sbcC*, and *hsdR* genes. Moreover, findings showing that λ Red markedly increased the efficiency of HR in *E. coli* [9] and led to the development of tools for genetic modification and their use in *E. coli* and other microorganisms [10,11,44]. The method illustrated in Figure 1A requires only 35 bp of homology for efficient recombination and was adopted by the Japan genome project to construct the Keio collection of single-gene *E. coli* deletions [10,45]. Greater understanding of the molecular mechanism of λ Red HR led to many improvements for its use in *E. coli*, phages, and other bacteria [46,47,48,49,50]. Targeted replacement by HR requires suppression or inhibition of the RecBCD exonuclease. Historically, the mutants used had reduced RecBCD activity and a mutation in *sbcA* for activation of the *recET* pathway, which together leads to HR by the RecET recombinase. Major breakthroughs came from utilizing λ Red, which encodes the *gam*, *bet*, and *exo* genes, for recombination of PCR products with short homologies flanking the chromosomal target [10,11].

The original method employs an antibiotic resistance fragment carrying a resistance gene flanked by FLP recognition target (FRT) sites, which allow elimination of the resistance cassette with an FLP expression plasmid, leaving behind “scar” sequences. A strategy was subsequently developed for constructing “markerless” (scarless) gene replacements (deletions or substitutions) [56]. This strategy involved the introduction of I-SceI sites into the *E. coli* chromosome, which had been absent from the genome. This scarless protocol was subsequently combined with λ Red HR to create scarless deletions (Figure 1(B1)).

Figure 1(B2) illustrates an alternative protocol for constructing scarless mutations. Unwanted sequences may be removed from genomes using a counter-selection method. A killing gene, also called a suicide gene, which can provide strong negative selection, can be introduced into cells under permissive conditions when the gene product is inactive. Treating cells with an inducer of its synthesis or a compound that inhibits cells harboring its product provides selection against cells bearing the inhibitor gene. This method has been used to target different regions of the *E. coli* chromosome, including *sacB* [57,58,59], *tetA* [58,59,60], and *tolC* [61,62,63]. The leakiness of these genes, however, often prevents their use as general tools in genomics studies, requiring the implementation of both positive and negative selection with a dual *tetA-sacB* cassette [58] or an optimization protocol to interfere with escape [63]. Two-step protocols in Figure 1(B2) have provided powerful tools for recombineering *E. coli* and related bacteria with single-stranded (ss) and double-stranded (ds) DNA [31,64,65,66,67,68,69,70,71,72,73].

### 2.3. Genome-Scale Modification in the Genome Research Era

#### 2.3.1. Deletion of a Large Genomic Region by Random Tn Insertion

Site-specific recombination at the *loxP* site resulted in two different types of Tn insertion mutations. Tn insertions located at both ends to be deleted were selected from each insertion mutation library and combined on one genome using P1 transduction. In the presence of overexpressed Cre protein, the fragment located between the two types of Tn was removed by Cre-*loxP* site-specific recombination (Figure 1C) [74]. These transposons were used to construct separate large-scale Tn insertion libraries, which were subsequently combined in the same strain by P1 transduction to yield a double Tn mutant. A site-specific recombinase was introduced to remove the chromosome segments between the Tn elements. This method has been used to construct large-scale genomic deletions by, for example, deleting 60 to 120 kb between pairs of Tn elements and choosing those mutants that did not impair cell growth. Cre was subsequently used to eliminate segments between *loxP* sites, followed by the introduction via P1 transduction of Tn elements for an additional large deletion and the repeating of the entire process (Figure 1(C,C1)).

Another approach for deleting large chromosomal segments randomly relies on a complex transposon with one Tn element inside another, Tn-in-Tn, carrying two types of Tn elements with opposite terminal repeat directions (Figure 1(C2)) [52]. Following random transposition of Tn-in-Tn into the chromosome, synthesis of the transposase for the internal element is induced, leading to its transposition to a new site and elimination of DNA between the original and new sites. When this process was repeated 20 times, the average deletion length was found to be about 10 kb, with a total of about 200 kb being successfully deleted [52]. In addition to Tn elements for making simple deletions, a Tn element carrying a conditional replication origin was constructed, allowing recovery of the deleted fragment as a plasmid. Therefore, this system may allow deletion of essential genes. To date, a comprehensive set of single-gene deletion mutants has been constructed [45,75], but it would be advantageous to simultaneously obtain clones of these fragments. Of the 15 cases examined, 11 were free of essential genes because growth was observed even when the plasmid was removed, whereas the other four were no longer viable after the plasmid was removed.

#### 2.3.2. Large-Scale Deletion by HR

*E. coli* is thought to have acquired many genes to survive in diverse environments. Shrinking the *E. coli* genome is thought to improve the efficiency of metabolic functions and reduce redundancy in genomic and regulatory structures [76,77]. Using HR, the *E. coli* K-12 genome lacking K-islands, which were identified by comparative genomics as recent horizontal acquisitions to the genome, was reduced (Figure 1(B1)) [16]. This method, which was based on the accumulation of scarless deletions by HR and DSB, allowed the elimination of 15% of the *E. coli* K-12 genome. Twelve K-islands, containing fragments of cryptic phage, transposons, disrupted pseudogenes, and genes of unknown function, were deleted. Ultimately, 9.3% of genes in the genome, including 24 of 44 transposon regions, were deleted [16]. Furthermore, strains with large deletions grew as well on minimal medium as wild-type strains, confirming that these K-islands did not contain essential genes.

These results suggest that mobile elements such as IS, which may drive evolution but induce genomic instability, can be deleted, as can genes with unnecessary function and groups of genes that adversely affect the bacterial growth environment, including in humans. However, it is not easy to predict which genes have those functions. A comparison of genomes of different *E. coli* strains enabled the selection of genes that were present in K-12 but absent in other *E. coli* strains. This resulted in the identification of a set of candidate genes, constituting about 20% of the genome, for deletion. This is an example of purposefully designed deletions that contain unstable factors and gene groups that are not necessary for bacterial growth. Moreover, the strains with large deletions, such as MDS42 and MDS43, grew almost as well as wild-type, with the stability of their genomes and their transformation efficiency being improved. 

Another study first compiled a list of predicted essential genes, followed by the use of an HR method to delete regions between these genes [53]. Using λ Red HR, alternate Ab resistance cassettes were inserted into intergenic regions between two essential genes, followed by combining them by P1 transduction and eliminating the inserted cassettes with λ Red HR. About 30% of the *E. coli* genome was deleted by combining the largest deletions between essential genes using P1 transduction, with the resulting phenotypes analyzed by determining their cell shape and nucleoid organization.

### 2.4. Genome-Scale Genetic Modification in Systems and Synthetic Biology

#### 2.4.1. CRISPR-Cas Application

Technological developments since the elucidation of the CRISPR-Cas mechanism have marked a major turning point in the biological sciences, just as the discovery of junctions, restriction enzymes, and vectors paved the way for molecular biology. CRISPR-Cas provides a process to design cleavage sites, called double-strand breaks (DSBs), at will. Figure 1(B1) shows how targeting the I-SceI site to a specific location generates a DSB that leads to the formation of the designed deletion. Figure 1(C3) shows how random Tn mutagenesis can be used to generate DSBs and nearly random deletions by CRISPR-Cas [78].

CRISPR-Cas is most often used to create DSBs at locations governed by the sequence of guide RNA (gRNA). Accumulation of DSBs if unrepaired is lethal. In eucaryotes, most broken DNA ends are bridged by nonhomologous end-joining (NHEJ) [79]. NHEJ, however, is usually not possible in bacteria due to a lack of the key NHEJ proteins Ku and Ligase-D. In *E. coli*, DSBs are most often repaired by HR and less frequently by alternative end-joining. An alternative end-joining (A-EJ) mechanism of repairing DSBs involves end-resection by RecBCD, end synapsis via microhomologies, and ligation of DNA ends [80,81] by LigA (Figure 1(C3)). Combining CRISPR-Cas with λ Red HR permits fashioning scarless deletions, insertions, or substitutions by design, limited only by occurrences of PAM sequences (Figure 1(D1)) [82,83,84,85,86].

The development of mutant Cas proteins lacking endonuclease activity has allowed precise base editing at very limited target sites in genomes by fusion of the enzyme cytosine deaminase to an inactive Cas subunit [87,88,89,90,91,92], and various point mutations and small insertions/deletions by fusion of the reverse transcriptase to a single active Cas (nickase) subunit [93,94,95,96] (Figure 1(D2)). This subject has previously been reviewed [97].

The generation of catalytically inactive Cas by mutation has allowed repurposing CRISPR as an RNA-guided platform that can specifically interfere with transcription elongation, RNA polymerase binding, or transcription factor binding (CRISPR interference; CRISPRi) using a single guide RNA (sgRNA) chimera [98,99,100,101,102].

CRISPRi was used to analyze a group of essential genes in *B. subtilis*, although this approach did not focus directly on genome modification [99]. CRISPRi screening of *E. coli* was performed by synthesizing a library of 92,000 sgRNA sequences covering the entire genome, with PAM sequences as the only constraint [100], thus identifying *E. coli* essential genes and genes essential for phage λ growth [101]. In a separate study, 60,000 sgRNAs were evaluated to test essentiality while also assessing the design of sgRNAs for all genes, including those that did not encode RNA [102]. Essentiality was tested with a pooled library, with the results evaluated by determining the relative change in read count by next-generation sequencing (NGS). The rules for effective gRNA design have been summarized [100,102].

#### 2.4.2. Acceleration of Evolution under the Constraint of Mutation Direction by Oligo DNA

The 1990s marked a turning point in biological research, beginning with the automation of DNA sequencing and the development of technologies for the production of large amounts of data [103]. At the start of the 21st century, the pace of development of many technological innovations continued to increase. Genome modification using HR has been based on methods of design and recombination. However, the mutation of many genes simultaneously can result in the synthetic lethality of genetic interactions, in which one mutation affects other genes, making the accumulation of mutations difficult. The relatively few analyses of genetic interactions have made it difficult to design methods that take these relationships into account.

Therefore, a method was devised to accelerate the evolution of mutagenesis by adding the constraint of viability and utilizing the principle of HR of λ Red, while restricting sequences using synthetic DNA. This Multiplex Automatable Genome Engineering (MAGE) method uses the β protein of λ Red and long (90 nt) synthetic ssDNAs, allowing acquisition of mutations on a genome-scale without lethal or severe growth-defect mutations because mutant selection is based on cell growth (Figure 1G) [54]. Although MAGE was originally performed robotically [54], it can also be performed manually. MAGE has provided a powerful tool for genome-wide codon replacement [104], metabolic engineering [105], and other biological functions (Figure 1G) [106]. λ Red HR is increased in mismatch repair mutants and shows strand bias [55,107]. MAGE technology has been further improved by using MAGE and λ Red HR to stimulate the evolution of host *E. coli* primase and helicase [108], which control the length of Okazaki lagging-strand fragments [109].

### 2.5. Impact of Plasmid Clones on Genome-Scale Analysis and Resource Construction

The construction of plasmid libraries encoding genes is important for understanding gene function. Resources required before starting a genome project include plasmid clones and mutants of the target gene, making their construction the first step. Once construction is underway and a clear blueprint of the target organism has been obtained, it is desirable to have systems to analyze the organism in its entirety. High-density DNA membrane filters and microarrays for *E. coli* have made global analysis possible [110,111,112,113]. Once all the gene clones and mutant libraries have been established, complete global comparative analysis of the entire gene set under the same conditions is feasible.

These requirements have given rise to global resource-building activities, and the construction of plasmid clone and mutant strain libraries [45,114,115,116], including promotor libraries [117]. The plasmid clone library was started using the restriction enzymes cloning method rather than recombination. Development of many of the methods described below allowed editing of the *E. coli* genome. Cloning of individual genes allowed the creation of synthetic metabolic pathways, fine-tuning of promoters, and the alteration of codons, after which the genes were recombined into the genome. These protocols not only optimize synthesis of useful products but modify *E. coli* biosynthetic and catabolic pathways. For example, *E. coli* genes for glycolysis are scattered throughout the genome. Modulating expression of these genes requires bringing them together [118] by using Ordered Gene Assembly in *Bacillus subtilis* (OGAB) [119] to create a series of plasmids with genes of the *E. coli* glycolytic pathway clustered in a variety of arrangements.

The cloning of synthetic DNA fragments into plasmids for growth in *E. coli*, other bacteria, and yeast is a key step in the chemical synthesis of genomes [120]. Methods using HR have been developed, including commercially available In-Fusion [121], SLiCE [122], and Gibson Assembly [123] methods. These technologies, which are often available as kits, have made genomic manipulations easier and faster, and have contributed to genome modification as the first step for cloning target genes.

### 2.6. Accumulation of Gene Modifications on One Genome

Once each of the regional genomic modification has been created, it is often important to transfer mutations resulting in antibiotic resistance to other strains, thereby aiding the functional analysis or accumulation of mutations on one genome. Three convenient methods of mutation transfer are currently available to transfer mutations into the target host strain: (1) conjugation, (2) P1 transduction, and (3) PCR fragmentation (Figure 2). All of these methods create mutants by HR.

Historically, most mutations are detected by selection markers, with one example being the antibiotic-resistant template Flp-FRT (Figure 2) [10]. Other markers can also be used if they can be selected. In addition, site-specific recombination between FRTs can be significantly altered by changing the spacer sequences inside the FRTs [124]. By contrast, CRISPR enables markerless accumulation of mutations using PCR fragments (Figure 1(D1)).

## 3. Genome-Scale Projects

### 3.1. Minimal Genome Projects

The minimal genome project, which began in 1997 based on a concept of minimal gene sets [125,126], has been ongoing for many years but has not yet been completed. The main reasons for non-completion are the existence of alternative pathways and compensatory circuits in the intracellular functional network, and the existence of orphan enzymes whose genes have not yet been identified. The minimal genome concept was later linked to the concept of chassis genomes in synthetic biology [127].

#### 3.1.1. Large-Scale Deletion by Random Transposon Insertion

Strains with large-scale deletions have been constructed using *loxP* site-specific recombination sites embedded in two types of Tn5 and a fragment of Tn5 randomly inserted into the *E. coli* genome [74]. For example, two types of Tn5 incorporating separate Kan and Chl antibiotic resistance genes and *loxP* site-specific recombination sites were constructed and used to generate random insertion mutant libraries. Mutants were selected from each Kan- and Chl-resistant library that flanked the region to be deleted, with the two insertion mutations combined into a single genome by P1 transduction. Addition of the Cre protein allowed the large-scale deletion of the genomic region between the *loxP* site-specific recombination sites. This method was used to introduce large-scale genomic deletions in six strains by combining two types of transposons, deleting 60 to 120 kb between them, and selecting mutations that did not impair the growth of the deleted cells. The selection markers between *loxP* sites were subsequently removed from the strains by Cre, followed by the introduction of another large deletion region into a single genome by P1 transduction, resulting in the construction of a minimal genome.

By contrast, Tn5 derivatives have been used to create a second transposon transposition from within the transposon once inserted into the genome by adding another set of transposon recombination sites. Repeating these deletions resulted in a minimal genome [52]. Because the average length of each deleted region was about 10 kb, repeating this process 20 times successfully introduced deletions into a ~200 kb region. In addition to designing, a Tn carrying a condition-sensitive replication origin was designed to rescue as a plasmid the fragment to be deleted in the deletion strain. This system may allow the introduction of deletions into essential gene regions. A comprehensive single-gene deletion strain has been constructed, and the essential genes, as shown by single deletion, have been identified [45,75]. The essentiality of large-scale deletions can be analyzed by simultaneously obtaining clones of these fragments. Of the 15 cases analyzed, 11 lacked the essential genes because growth was observed even when the plasmid was removed. By contrast, the other four cases were no longer viable after removal of the plasmid. One case contained an essential gene, whereas the others contained deletions of short regions that did not contain any ORFs, making their situation unstable. The number of copies of the plasmid, however, may be a factor.

#### 3.1.2. Large Scarless Deletion by HR

The reduced genome lacking K-islands [16] has been further improved [76]. Although *E. coli* was originally described as an intestinal bacterium, it has acquired a diverse set of genes, enabling it to survive in various environments. Shrinking of the genome is thought to improve the efficiency of metabolic functions and reduce redundancy in genomic and regulatory structures [76]. Mobile elements, such as IS, which may drive evolution but induce genomic instability, genes with unnecessary function, and groups of genes that adversely affect the bacterial growth environment, including in humans, can be deleted. However, predicting the genes that have these functions is difficult. A comparison of the genomes of different *E. coli* strains identified selected genes that were present in K-12 but absent from other *E. coli* strains, resulting in the selection of a set of candidate genes, comprising about 20% of the genome, for deletion.

The deletion method [16] was based on the accumulation of scarless deletions by HR and DSB, resulting in the successful deletion of 15% of the *E. coli* K-12 genome. This was an example of purposefully designed deletions of sequences that are unstable factors and gene groups that are not necessary for the growth of *E. coli*.

The growth of the final strains with large deletions, MDS42 and MDS43, were almost the same as that of the wild type, although these deletions improved genome stability and transformation efficiency, making these strains a practical, reduced-genome *E. coli*. MDS69, an improved *E. coli* strain with additional deletions, which is currently available commercially from Scarab Genomics (https://www.scarabgenomics.com/products/clean-genome-e-coli/, accessed on 4 September 2022).

A method of scarless deletion of a region of non-essential genes between essential genes involved the performance of two HR events (Figure 1(B2)) [53]. This method resulted in the deletion of the largest possible region from the essential intergenic region and the accumulation of deletions by P1 transduction to yield a minimal genome with large deletions.

Analyses of the phenotypes of *E. coli* from which about 30% of the genome had been deleted showed that the growth rate was inversely associated with the size of the deleted region [53]. Deletion also altered cell morphology, with changes in cell length and width and in nucleoid organization. Attempts to combine these large deletions showed that some could not be combined (Kato, J., personal communication), perhaps because combination resulted in synthetic lethality. Therefore, it is still difficult to determine the associations between combinations of gene deletions and specific phenotypes. This deletion project has since become a joint project with Kyowa Hakko Co., Ltd, Machida, Japan.

Efforts have been made to develop bacteria with beneficial genomes for the production of materials, especially with industrial applications, without the inhibition of cell growth. One strain, MGF-01, was generated by deleting 1.03 Mb from 53 regions using P1 [128]. These deletions increased glucose consumption 1.44-fold and acetate accumulation 0.09-fold, confirming the efficacy of this method [128].

The strain MS56 has a genome reduced by 23% [129]. It was generated by removing IS and other factors that may cause instability in plasmids containing foreign genes, and its stability and efficiency of expression of foreign gene products was analyzed [129]. Evaluation of the scarless HR deletion method [16,76] with human tumor necrosis factor-related apoptosis-inducing ligand (TRAIL) and bone morphogenetic protein-2 (BMP2) showed its superiority. This led to the development of genome-reduced strains useful for biosynthesis in industrial applications.

### 3.2. Large-Scale Genome Modification by Synthesis and Recombineering

#### 3.2.1. Recoding Genome by Recombineering and ssDNA Accelerated Evolution

Despite the significant progress of the minimal genome project in accumulating individual genes with specific functions, it is still almost impossible to fully design a genome with deletion of many genes. These drawbacks may be overcome by genetic interaction analysis of systematic double deletion strains.

Evolutionary methods have also been explored. Mutants that are not viable or grow very slowly are eliminated during the selection process. Clarification of the molecular mechanism of λ Red recombination has shown that ssDNA predominantly introduces mutations into the replicating lagging strand through the activity of the beta protein alone [46]. These findings led to the development of the Multiplex Automated Genome Engineering (MAGE) method using in vivo evolution with multiple types of designed ssDNA and β proteins (Figure 1G) [54]. A homemade automated facility was also developed to automate this cycle, resulting in the mutation in a single step of 24 genes in the 1-deoxy-D-xylulose-5-phosphate (DXP) biosynthesis pathway scattered throughout the genome in a single step. Evaluation of the resulting mutations for optimization of DXP synthesis showed that, for 20 of the 24 genes, 90-nucleotide long ssDNAs were designed to optimize the ribosome binding site, increasing their levels of expression. For the four other genes, ssDNAs were designed to introduce nonsense codons, making them non-functional in the MAGE method. The running of 5–35 MAGE cycles resulted in ~10^5^ mutant strains and increased the production of the target product, isoprenoid lycopen, up to 5-fold within 3 days [54].

This development has enabled genome modification by deliberately limiting the direction of mutation and accelerating evolution by recombination of many parts of the genome at once. This technology has since been further improved, allowing its use on a larger scale. For example, a recoding genome was constructed by replacing the TAG termination codons on all 314 *E. coli* genes bearing these codons with TAA termination codons [104]. Because these 314 genes are scattered throughout the genome, the genome was divided into 32 regions, with the MAGE cycle run for each region to obtain evolutionary mutant strains. The Conjugative Assembly Genome Engineering (CAGE) method was also developed to integrate the mutated chromosomal sites into a single *E. coli* strain using conjugation, resulting in an *E. coli* strain in which all terminal TAGs were replaced by TAAs. Although recoding was expected to eliminate the need for TAG codons, *prfA*, the gene encoding releasing factor (RF1), which recognizes TAG codons, was deleted, shows the ability to replace TAG codons with other codons. This technology was further improved by developing primase and helicase mutant strains, which contain mutations that control the lengths of Okazaki fragments synthesized by the lagging strand [108].

It may also be possible to replace codons for a specific amino acid, rather than termination codons. Forty-two highly expressed essential genes were selected and rare codons in these genes were replaced by DNA synthesis; if this method was unsuccessful, these codons were replaced using MAGE. Ultimately, 405 codons on 42 highly expressed essential genes were replaced, resulting in reductions in cell growth [130]. These results showed that genome-wide codon replacement is feasible and that codons can be replaced using MAGE, with minimal or no effect on cell growth. Providing artificially modified organisms with a genetic code that does not exist in the natural world would thus ensure the safety of these organisms, even if they are released to the outside world.

#### 3.2.2. Recoding Genome by Synthesis

A method has been developed to replace a target region of the genome with a fragment of synthetic DNA designed for recoding by assembly in vivo, including recoding of the entire *E. coli* genome (Figure 1E). For example, replacement of the codons UAG (stop), AGG-AGA (Arg), AGC-AGU (Ser), and UUG-UUA (Leu) with other synonymous codons from the genome resulted in the construction of an *E. coli* genome with 57 codons. Similarly, replacing the codons TCG, TCA (Ser), and TAG (stop) resulted in the construction of an *E. coli* genome with 61 codons. Both methods used designed synthetic DNA, with the genome-reduced strain MDS42 used as the parent strain.

Although these methods showed some differences in their details, both involved assembly of the genome by HR in yeast cells and its transfer to *E. coli* cells. In one method, the target region on the genome was removed, the assembled recoded genome was inserted into the target region using *attL-attP* site-specific recombination, and the vector region was deleted by CRISPR [131]. In the other method, fragments that accumulated on the BAC vector were integrated into the recipient genome by transferring them to the recipient using conjugation, although assembly in yeast cells was the same [132]. This method resulted in the construction of a partially recoded genome by linearizing both ends of the fragments that had accumulated on the vector by CRISPR double-strand breaks and replacing them with the target regions of the genome by HR using λ Red recombinase. This step was repeated to construct the entire recoded genome [133]. The resulting *E. coli* strain Syn61 with a recoded genome was found to grow more slowly and have a longer cell length than the parental strain MDS42.

Only two assembled fragments significantly affected cell growth [131]. The responsible genes were identified and individually modified to overcome this drawback. One gene was found to be insufficiently expressed in the fatty acid biosynthesis operon *rpmF-accC*, an insufficiency circumvented by improving the promoter in the duplicated region using the MAGE method [131].

### 3.3. Resources for Genome-Scale Functional Analysis towards Genome Design

The genomic sequences of the *E. coli* K-12 strains, MG1655 and W3110, were compared to more accurately determine their sequences [134] and annotation [135]. Experimental resources were also designed to analyze *E. coli* genes globally. The initial focus was on construction of an ORF plasmid clone library with PCR-amplified genes from predicted ORF regions within the genome (Figure 3) [115]. Full-length cDNA microarrays were developed and shared with the research community to launch OMICS research [110,111]. Efforts were also made to construct gene deletions using the Kohara λ phage ordered clone library of the *E. coli* K-12 genome [136]. However, HR with synthesizable base length was problematic at that time. Immediately after λ Red HR was first used to disrupt genes on the *E. coli* chromosome with PCR products [10], a comprehensive library of *E. coli* single-gene deletion mutants, the Keio collection [45], was constructed, with this library made freely available to the research community as an open resource.

The Keio collection was used to examine the effects of central metabolic pathway gene deletions on transcription, translation, and intracellular metabolites levels [137]. Quantitative fluctuations of metabolites were small and remained stable compared with transcription alterations. Robustness was further addressed by analyzing comprehensive synthetic lethality through double gene knockout. This was accomplished by developing a second library of single-gene deletion mutants, the ASKA barcode deletion collection, which has not yet been completed. In addition to changing the antibiotic resistance gene, this library carried a 20 nt random sequence as a barcode (Figure 4) [138].

To date, two independent representatives of about 3000 genes have been successfully isolated, confirmed, and stored. Each of the independent representatives of the same gene deletion has a unique barcode. A method was also developed to efficiently produce double gene deletion strains by combining two types of deletion strains and measuring their growth, thereby enabling analyses of genetic interactions [139,140].

Catalytically inactive Cas resulting from gene mutation has allowed repurposing CRISPR as an RNA-guided platform that can specifically interfere with transcription elongation, RNA polymerase binding, or transcription factor binding (CRISPR interference; CRISPRi) by using a single guide RNA (sgRNA) chimera [98,99,100,101,102]. CRISPRi has been utilized to analyze a group of essential genes in *B. subtilis*, although this approach did not focus directly on genome modification but on knockdown of gene expression [99]. *E. coli* was subjected to CRISPRi screening by synthesizing a library of 92,000 sgRNA sequences covering the entire genome randomly, with PAM sequences as the only constraint [100]. This enabled identification of *E. coli* essential genes and genes essential for phage λ growth [101]. In a separate study, 60,000 sgRNAs were evaluated for testing essentiality while assessing the design of sgRNAs for all genes, including non-coding RNA genes [102]. Essentially was tested using a pooled library, with the results evaluated by determining the relative change in read count by NGS. The rules for effective gRNA design have been described [100,102].

### 3.4. Genome-Scale Analysis towards Genome Design Platform

In genomics, the construction of mutants is an important first step in analyzing the biological functions of genes and their products. As of September 2020, the *E. coli* genome, annotated as GenBank entry U00096.3, included 4609 genes, with 4285 of these genes encoding proteins, many of which have unknown functions. In addition, this genome was found to include genes encoding small proteins and non-coding RNAs [141]. Genome-scale metabolic models of *E. coli* have been developed, such as iJE660 [19], iJR904 [20], iAF1260 [21], iJO1366 [22], and iML1515 [23], with others still being developed and improved. Refinements of these models have shown the presence of as yet unidentified alternative pathways and isozymes and gaps in metabolic networks (orphan reactions) [24,142]. Knowledge of *E. coli* is also incomplete [22,24,142].

The minimal gene set concept [125] has become the minimal genome project, but it is still far from complete. By contrast, the minimal genome concept has expanded to the concept of a minimal genome factory that optimizes the genome to produce valuable products [143]. These concepts have now expanded to include the concept of the chassis genome in synthetic biology [127,144].

Evaluation of *E. coli* identified 325 genes that could not be singly deleted [45,75]. However, even non-essential genes can be lethal in combination with other genes, making them synthetic lethal genes. Epistasis or interactions between genes and mutations are important for understanding gene function in *E. coli* and other cells [139,140,145,146,147,148,149].

Gene combinations of alternative pathways, compensatory pathways, and isozymes often exhibit synthetic lethality or synthetic sickness. Such genetic interactions can provide new insights into gene function [99,139,140,150,151]. Advances in genome design or genome deletions by design will require the systematization of knowledge from many sources, computer models for design, and genome editing technologies to enable experimental validation [152].

## 4. Genomes by Synthesis

Although, to our knowledge, the *E. coli* genome has not been chemically synthesized, a genome has been reconstructed using completely synthetic DNA and *Mycoplasma* cells (Figure 1F) [17,18]. The genomic sequence of a living bacterium that serves as a template is required, although, in the near future, genomes may be reconstructed using fully synthetic DNA, based on the genome sequence design in *E. coli*. Determination of the rules for genome construction is absolutely required.

## 5. Discussion and Perspective

### 5.1. Transition of Biological Concepts

About 50 years have elapsed since recombinant *E. coli* gene modification technology was initially introduced, from the elucidation of its molecular mechanism to the development of methods based on phage recombination mechanisms and improvement of the technology. This has enabled almost any type of sequence modification, from genome-scale large modification to base-level modification. Genome research in the 1990s may therefore represent a turning point in biology, both technologically and conceptually. This situation was similar in the 1970s, when genetic modification techniques were developed and molecular biology began to make significant progress.

Often, new concepts are not immediately accepted. One such example is the difference between “forward genetics” and “reverse genetics”, which was initially developed by physicists but was not fully accepted by researchers in genetics (Yura, T., personal communication). Differences in acceptance were not likely due to differences in ways of thinking, but increased understanding was likely due to experimental efforts. The throughput of sequencing technology has expanded about 1000-fold from the start of the project in 1990 to the completion of the first draft sequence of the human genome. With the technology available at that time, the genome of *E. coli* took 7 years to complete. However, the 21st century has seen the development of sequencing technologies based on novel concepts, and sequencing is now more than a billion times more efficient than it was in 1990. The driving force behind this development dates back to the 1990s. Researchers understood the importance of comparative analysis of individual human genomes and the need for further development of sequencing technology as the next step after the completion of the human genome.

Technological innovation has not only affected the speed of analysis, but has made possible more precise and diverse analyses, and at lower cost. For example, the range of applications of sequencing technology has rapidly expanded to include analyses of gene expression, protein–DNA interactions, protein–protein interactions, nucleoids, and species distribution in populations. Moreover, genome editing in the 21st century has been revolutionized by determining the molecular mechanism underlying CRISPR.

Although the recombination efficiency of *E. coli* is lower than that of other model microorganisms, such as *B. subtilis* and yeast, the tooling of λ Red recombinase made possible the modification of *E. coli* by genome-scale recombination, increasing its recombination efficiency. The availability of genome modification techniques by recombination in Gram-positive, Gram-negative, and eukaryotic unicellular organisms has created a favorable research environment for comparative analysis.

Progress in genome analysis led to the development of research resources in yeast, the creation of databases, the construction of gene clone and deletion strain libraries, and their sharing as community assets [153,154]. For example, participating institutions in Europe and Japan worked together as a community to build a *B. subtilis* deletion strain library [155,156].

Although a similar comprehensive experimental resource community has been proposed for *E. coli*, few laboratories agreed to participate. Groups at Keio University, the Nara Institute of Science and Technology, and Purdue University therefore agreed to develop this resource. Although the time to completion was undetermined, the development of many innovations resulted in the completion of the entire project in 3 years. These developed resources were subsequently shared with the research community as open resources. *E. coli* could therefore be positioned as a model microorganism alongside *B. subtilis* and yeast.

The importance of putting all the pieces together is enormous. The importance of analyzing the structure and function of target genes and proteins in detail through individually targeted analyses will likely remain unchanged. However, looking at the entire picture revealed aspects that could not be determined from individual studies of a narrow range of targets. Molecular biology in the 20th century has been described as a very precise “science of parts”, whereas genome research, starting in the 1990s and extending into the 21st century, can be described as “science as a system”. Alignment of the two will likely greatly advance our understanding of the whole picture of life.

### 5.2. Concept of Minimal Genome

The minimal genome project, which started with the minimal gene set, has been a long-term effort to realize the minimal genome concept. Initially, the direction of the project was purely biological: to construct the minimum genome necessary for cell growth in nutrient media. Approaches included the removal of large areas of the genome that could be removed [74,157] and the removal by design of areas considered unnecessary for cell growth (see Section 2.5) [76]. In particular, the large-scale deletion efforts showed that, although deletion of individual genes in a region did not significantly affect the growth of an organism, simultaneous deletion of many gene clusters significantly affects growth and may even be lethal. These findings emphasized the importance of determining genetic interactions under conditions of synthetic lethality and sickness, and may have a major impact on network biology. Indeed, findings showing that cell lethality was due to multiple gene deletions, with quantitative analyses of the effects of single-gene deletions on intracellular conditions [137] suggesting that many genes that are considered non-essential may repeatedly interact with each other to maintain intracellular stability, at least at the metabolite level. Efforts are underway to analyze genetic interactions through the systematic construction of double gene deletion strains. The saying in Japan, “If the wind blows, the bucket makers prosper”, is equivalent to the “Butterfly Effect” in chaos theory. Cells dynamically regulate transcription, translation, and enzyme activities while optimizing the balance between the genetic elements of the cell and the environmental factors affecting growth.

The minimal genome concept has also expanded to optimize the practical use of cells for industrial purposes [129,143]. Moreover, the beginning of synthetic biology has resulted in expansion of the concept of the “chassis genome” [127,144].

Completely designing a genome is still not possible. This possibility may be enhanced by accumulating and analyzing genetic interaction information, including all genetic interactions and trans-omics analyses.

### 5.3. New Research Targets and Fields

The deletion of a single gene can affect the expression and function of a group of related genes. Although the individual effects may be small, the overall effect can be significant. Methods that look at the whole picture, such as RNA-seq, have made possible the analysis of how these initial disturbances affect the whole. The availability of comprehensive research resources has made possible the comparisons of the phenotypes associated with all gene deletions. In combination with other analytic methods, phenotypic analyses have made multidimensional analyses possible, as well as assessments of the position of individual gene groups in the overall activity of the cell. Thus, it has become experimentally possible to view cellular activity as a network [158].

Research on *E. coli* has followed this trend, with *E. coli* being one of the model organisms leading the new biology of the 21st century. This has led to the development and utilization of comprehensive research resources, including plasmid clones [115,116,159], deletion strains [43], and promoter fragment clones [117]. Several other resources remain under development.

### 5.4. Dramatic Changes in Quality and Quantity of Data Require a Variety of Analyses

The existence of comprehensive research resources and the development of analytic methods have led to a rapid increase in data. Effective use of these data requires mathematical analyses and information processing technologies such as statistical analysis and modeling. Although analyzing the data generated from a single comprehensive study provides many hints on physiological functions and molecular mechanisms, these hints may not be successfully verified by experimental methods. Although data registered in databases and accessible through Web systems have been used for informatics analysis, the contribution of these enormous amounts of data to the progress of individual research is unclear. The situation may be dependent on the presence of a framework for sharing information that may provide direct material for experimental validation, such as proposals for individual molecular mechanisms obtained from informatics analysis. Another factor may result from the same research group performing comprehensive analyses and individual targeted research. The accumulation of experience is important in deepening research in individual studies, making it difficult to complete both comprehensive and individual analyses in a single laboratory. Establishment of a community-oriented framework is likely necessary to share comprehensive data, along with interpretations and/or suggestions. For example, a group at McMaster University in Canada has specialized in discovering antimicrobial antibiotics and has developed research using comprehensive resources. As soon as resources become available, they are used for exhaustive screening [160], leading to the acceleration of research and development, and further expansion [161]. This group is therefore a good example of the successful use of both comprehensive and individual approaches in a single laboratory.

## 6. Epilog

The flow of genome modification in *E. coli* has increased expectations that researchers will have sufficient knowledge and technology to design organisms. To date, however, the knowledge to design genomes is insufficient. However, the steady accumulation of knowledge, evolving technologies, and progress in modeling has increased the understanding of genome organization. The ability to completely design artificial microbial cells may enable the construction of cells that can be used in medicine, antibiotic discovery, and engineering.

## Figures and Tables

**Figure 1 microorganisms-10-01835-f001:**
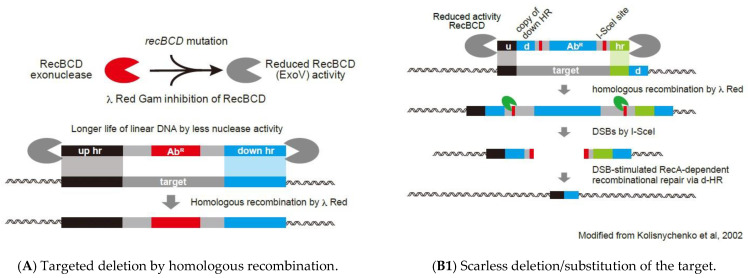
Schematic illustration of genome-scale modification methods. (**A**) Two methods have been used to block RecD exonuclease: (1) using *recBC* mutations and (2) λ Red Gam synthesis. Cells are transformed with linear double-stranded (ds) DNA encoding an antibiotic resistance (Ab^R^) cassette and ends at homology regions (hr) of upstream (up) and downstream (down) regions of the target. (**B1**) Scarless deletion using I-SceI nuclease. The drug-resistant fragment flanked by I-SceI restriction enzyme sites is amplified with up and next to the downside homology regions (u and hr) and introduced into the genome by λ Red homologous recombination [16]. The I-SceI-flanked segment is eliminated by expressing the meganuclease I-SceI, resulting in a double-strand break (DSB), DSB-stimulated DNA repair, and RecA-dependent recombination between the d direct repeats. This figure is modified from Kolisnychenko et al. [16] (**B2**) The Ab (antibiotic resistance gene) with the killing gene, such as *sacB*, *ccdB*, *parE*, or phage T7 0.7, under the control of the tightly regulated promoter, such as *rhaBp* [51], is amplified with 36 to 40 nt homology region (hr) to target. The amplified fragment is then transformed into a strain expressing λ Red to insert into the genome. Double-stranded “Substitution Sequence (SS)” with flanking hrs is transformed into the Ab-resistant fragment integrated strain expressing λ Red. The transformants are selected in the presence of L-rhamnose, preferably with L-rhamnose as the sole carbon source. (**C**) Random insertion mutagenesis by Tn. (**C1**) Mutant Tns with less sequence specificity for insertion sites on chromosomes have been developed, with transposons such as Tn5, Tn10, and Mariner often used. Two different Tns of two different drug resistance genes were randomly mutated, with the location of insertion on the genome determined by PCR and sequencing. *E. coli* strains with insertions at appropriate positions were selected and combined into a single *E. coli* strain using the P1 transduction method. This method used site-specific recombination at each Tn and induced recombination by increasing the production of site-specific recombinase and deleting the region between Tns. (**C2**) A complex with Tnp that recognizes IE is introduced into the cell to obtain the first random insertion mutation. Synthetic induction of Tnp recognizing the internal ME is then performed to obtain a transition mutation; the direction of the second transition results in a deletion between two different Tn insertion sites. This figure is modified from Goryshin et al. [52] (**C3**) Insertion of a Tn fragment into the genome, followed by CRISPR-Cas cutting of the inside of the Tn fragment. This yielded a strain in which nuclease activity deleted the periphery. (**D1**) DSB was induced by CRISPR-Cas, with DNA fragments transformed by bridging homologous regions at both ends of each double-strand break, resulting in genome repair and yielding to the circular genome. (**D2**) Genome editing by fusion protein with a function different from that of Cas protein. Left panel: fusion of cytosine deaminase to a Cas protein with mutation-inactivated DNase activities [53]. Right panel: fusion of reverse transcriptase to a Cas, which inactivates only the nick on the other strand, providing a template for repairing the nick site and introducing the mutation by a reverse-transcribed sequence 77. (**E**) Introduction of synthetic DNA fragments into cells, generally yeast cells, resulting in assembly of the fragments by in vivo homologous recombination. After the assembly, the assembled fragment is collected and transformed into *E. coli* cell and replacement of the target region in the λ Red-induced strain by homologous recombination. (**F**) Assembly of the synthetic DNA fragments in the cell, followed by circularization to reconstruct the genome. The synthetic genome was subsequently transferred to bacterial cells by cell fusion [17,18]. This figure is summarized from Gibson et al. [17] (**G**) Introduction of an ssDNA about 90 bp in length to be mutated in the cell via the induction of λ Red β protein, which promoted the introduction of mutations on the lagging strand during DNA replication and accelerated the introduction of mutations throughout the genome. This figure is modified from Wang et al. [54] and Costantino and Court [55].

**Figure 2 microorganisms-10-01835-f002:**
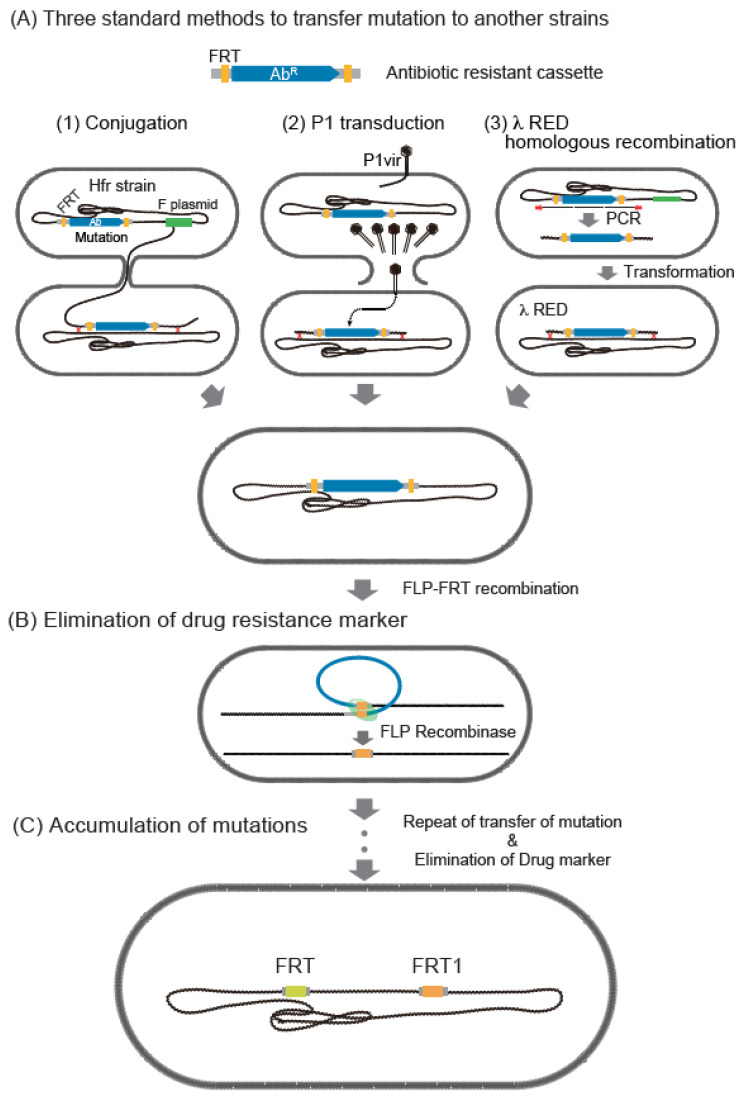
Common methods to transfer mutations to another strain using Keio collection mutants. (**A**) Methods of transferring mutations into the target host strain. (**1**) Conjugation, consisting of the recombination of an oriT onto the chromosome and use of an F plasmid to provide conjugative transfer factors in trans. (**2**) P1 transduction, using phage P1 lysate prepared on the mutant to infect new strains. (**3**) λ Red homologous recombination. (**B**) Elimination of drug-resistant marker by Flp-FRT site-specific recombination. After elimination, one copy of a 34 bp FRT scar remained. (**C**) Repetition of steps (**A**,**B**), resulting in the accumulation of mutations with FRT copies as a scar.

**Figure 3 microorganisms-10-01835-f003:**
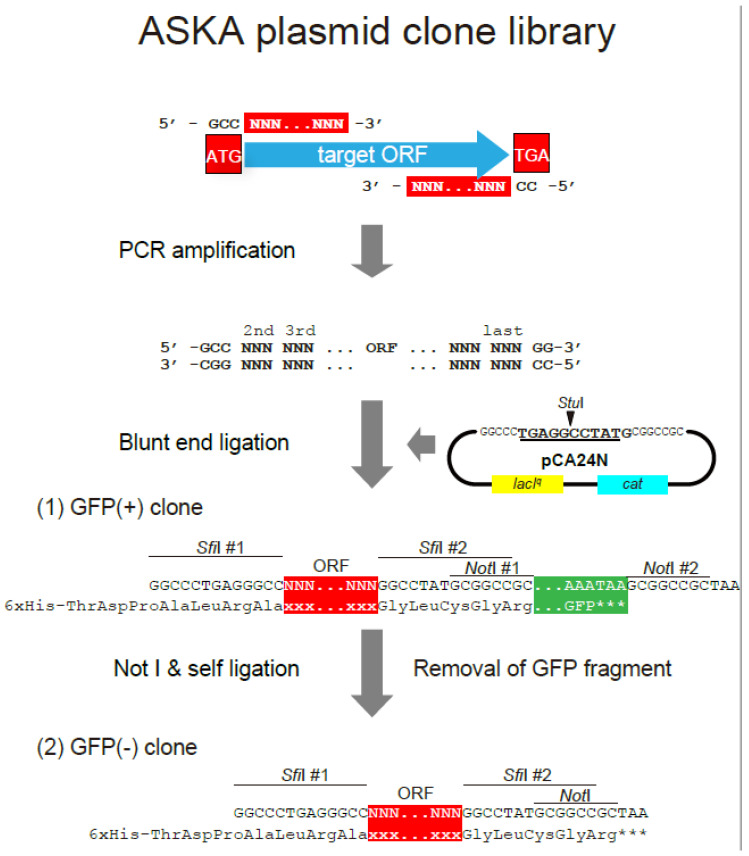
Construction of the ASKA plasmid clone library. The sequence corresponding to all of the amino acids in the coding region, except for the first Met codon, were PCR amplified, with additional GCC and CC nucleotides at the N- and C-termini, respectively. Translation from termination codon is shown by ***. The amplified fragments were subsequently cloned into the StuI site of pCA24N. Only clones with predicted orientation could generate fluorescence from the eGFP peptide. After the structures of the cloned plasmids were validated, the plasmids were cut with NotI and self-ligated to eliminate eGFP. The structures are of (**1**) a fusion type with eGFP and (**2**) a non-fusion type plasmid clone.

**Figure 4 microorganisms-10-01835-f004:**
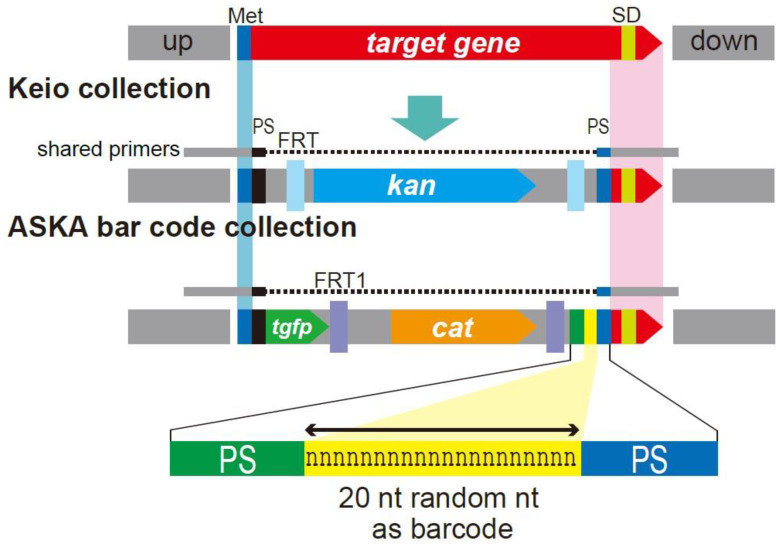
Structure of Keio single-gene deletion and ASKA barcode deletion collections. The coding regions of both deletion strains, except for the initiation codon and the codons encoding the six amino acids at the C-terminal, were replaced by drug resistance fragments. Site-directed recombination of FLP-FRT removed the drug resistance region. After removal from the Keio collection, the initiation codon and the codons encoding the C-terminal six-amino-acid region of the target gene were fused in frame with codons from the FRT site to suppress the polar effect of the downstream gene 35. The shared-primers contained the initiation codon (blue) and 50 bases upstream and downstream (gray), including the six C-terminal codons and the terminal codon (red) as chromosomal homologous regions. The black and blue sequences represent primers amplifying the template plasmid with the drug resistance gene and FRT sites both of Keio and ASKA barcode collections, respectively. For introducing 20 nt length random sequence as a barcode, PS for amplifying the resistant fragment (green), 20 nt random sequence (yellow), and PS of shared-primer (blue) was synthesized and amplified with PS (black) to prepare the template fragment. The resistant fragment with barcode was then amplified by shared-primers. The amplified drug resistance fragments for the Keio collection and ASKA barcode collection were used to transform λ Red-induced strains to generate deletion strains by homologous recombination. The barcode deletion strains are available for about 3000 genes.

**Table 1 microorganisms-10-01835-t001:** Recombination related genes.

ECK	Gene	Synonym	Left	Right	Ori	EcoCyc	UniProtKB	Description	Class	Origin
ECK2751	*casE*	*cas6e, cse3, ygcH*	2,873,696	2,874,295	C	G7426	Q46897	pre-CRISPR RNA endonuclease	nuclease	
ECK0232	*dinB*	*dinP*	247,385	248,440		G6115	Q47155	DNA polymerase IV	polymerase	
ECK0183	*dnaE*	*polC, sdgC*	201,613	205,095		EG10238	P10443	DNA polymerase III subunit α	polymerase	
ECK0215	*dnaQ*	*mutD*	232,554	233,285		EG10243	P03007	DNA polymerase III subunit ε	polymerase	
ECK0464	*dnaX*	*dnaZ*	488,097	490,028		EG10245	P06710-2	DNA polymerase III subunit γ	polymerase	
ECK0633	*holA*		666,579	667,610	C	EG11412	P28630	DNA polymerase III subunit δ	polymerase	
ECK1085	*holB*		1,151,767	1,152,771		EG11500	P28631	DNA polymerase III subunit δ’	polymerase	
ECK4252	*holC*		4,474,203	4,474,646	C	EG11413	P28905	DNA polymerase III subunit χ	polymerase	
ECK4363	*holD*		4,598,169	4,598,582		EG11414	P28632	DNA polymerase III subunit ψ	polymerase	
ECK1843	*holE*		1,919,914	1,920,144		EG11505	P0ABS8	DNA polymerase III subunit θ	polymerase	
ECK4339	*hsdM*	*hsm, hsp*	4,571,825	4,573,414	C	EG10458	P08957	Type I restriction enzyme EcoKI methylase subunit	others	
ECK4340	*hsdR*	*hsp, hsr*	4,573,615	4,577,127	C	EG10459	P08956	Type I restriction enzyme EcoKI endonuclease subunit	nuclease	
ECK4338	*hsdS*	*hsp, hss, rm*	4,570,434	4,571,828	C	EG10460	P05719	Type I restriction enzyme EcoKI specificity subunit	nuclease	
ECK1145	*mcrA*	*rglA*	1,206,351	1,207,184		EG10573	P24200	e14 prophage; Type IV methyl-directed restriction enzyme EcoKMcrA	nuclease	e14 prophage
ECK4336	*mcrB*	*rglB*	4,568,324	4,569,703	C	EG10574	P15005	McrB hexamer	nuclease	
ECK4335	*mcrC*		4,567,278	4,568,324	C	EG10575	P15006	Type IV methyl-directed restriction enzyme EcoKMcrBC subunit	nuclease	
ECK2152	*nfo*		2,244,868	2,245,725		EG10651	P0A6C1	endonuclease IV	nuclease	
ECK1144	*pinE*	*pin*	1,205,690	1,206,244		EG10737	P03014	Site-specific DNA recombinase of e14 prophage.	recombinase	e14 prophage
ECK1538	*pinQ*	*ydfL*	1,628,428	1,629,018		G6819	P77170	Predicted recombinase PinQ	recombinase	Qin prophage
ECK1369	*pinR*	*ynaD*	1,427,890	1,428,480	C	G6697	P0ADI0	Predicted site-specific recombinase	recombinase	Rac prophage
ECK3855	*polA*	*resA*	4,040,875	4,043,661		EG10746	P00582	DNA polymerase I	polymerase	
ECK0061	*polB*	*dinA*	63,429	65,780	C	EG10747	P21189	DNA polymerase II	polymerase	
ECK1348	*racC*	*sbcA*	1,412,294	1,412,569	C	EG10813	P15033	Rac prophage protein RacC		Rac prophage
ECK4381	*radA*	*sms*	4,616,278	4,617,660		EG11296	P24554	DNA recombination protein		
ECK1345	*ralA*	*ydaB, lar*	1,408,539	1,408,733	C	EG11900	P33229	endodeoxyribonuclease toxin RalR	nuclease	Rac prophage
ECK0883	*rarA*	*ycaJ, mgsA*	933,999	935,342		EG12690	P0AAZ4	Replication-associated recombination protein A.		
ECK2694	*recA*	*lexB, recH, rnmB, srf, tif, umuB, umuR, zab*	2,816,616	2,817,677	C	EG10823	P0A7G6	DNA recombination and repair protein; ssDNA-dependent ATPase; synaptase; ssDNA and dsDNA binding protein; ATP-dependent homologous DNA strand protein	recombinase	
ECK2816	*recB*	*ior, rorA*	2,946,369	2,949,911	C	EG10824	P08394	exodeoxyribonuclease V subunit RecB	nuclease	
ECK2818	*recC*		2,952,968	2,956,336	C	EG10825	P07648	exodeoxyribonuclease V subunit RecC	nuclease	
ECK2815	*recD*	*hopE*	2,944,543	2,946,369	C	EG10826	P04993	exodeoxyribonuclease V subunit RecD	nuclease	
ECK1347	*recE*	*rmuB, rac, sbcA*	1,409,592	1,412,192	C	EG10827	P15032	exonuclease VIII	nuclease	Rac prophage
ECK3692	*recF*	*uvrF*	3,874,057	3,875,130	C	EG10828	P0A7H0	Recombination mediator protein RecF		
ECK3642	*recG*	*radC, spoV*	3,819,119	3,821,200		EG10829	P24230	ATP-dependent DNA helicase	helicase	
ECK2887	*recJ*		3,030,281	3,032,014	C	EG10830	P21893	ssDNA-specific exonuclease	nuclease	
ECK3808	*recL*	*uvrD, uvrE, srjC, dar-2, dda, mutU, pdeB, rad*	3,991,892	3,994,054		EG11064	P03018	DNA helicase II	helicase	
ECK2612	*recN*	*radB*	2,745,703	2,747,364		EG10831	P05824	DNA repair protein RecN		
ECK2563	*recO*		2,695,649	2,696,377	C	EG10832	P0A7H3	Recombination mediator protein RecO		
ECK3816	*recQ*		3,999,773	4,001,602		EG10833	P15043	ATP-dependent DNA helicase RecQ	helicase	
ECK0466	*recR*		490,410	491,015		EG10834	P0A7H6	Recombination mediator protein RecR		
ECK1346	*recT*		1,408,790	1,409,599	C	EG11899	P33228	recombinase RecT	recombinase	Rac prophage
ECK2693	*recX*	*oraA*	2,816,047	2,816,547	C	EG12080	P33596	RecA inhibitor RecX		
ECK3826	*rmuC*	*dinK, sosB, yigN*	4,011,242	4,012,669		EG11472	P0AG71	DNA recombination protein		
ECK3398	*rpnA*	*yhgA*	3,537,075	3,537,953		EG11750	P31667	Recombination-promoting nuclease RpnA	transposase	transposon
ECK2299	*rpnB*	*yfcI*	2,416,677	2,417,567	C	G7197	P77768	Recombination-promoting nuclease RpnB	transposase	transposon
ECK0131	*rpnC*	*yadD*	143,455	144,357		EG11748	P31665	Recombination-promoting nuclease RpnC	transposase	transposon
ECK4329	*rpnD*	*yjiP, yjiQ*	4,559,364	4,560,284		G7934	P0DP21, P0DP22	Recombination-promoting nuclease RpnD	transposase	transposon
ECK2236	*rpnE*	*yfaD*	2,350,932	2,351,831		EG12323	P37014	Inactive recombination-promoting nuclease-like protein RpnE	transposase	transposon
ECK1862	*ruvA*		1,940,171	1,940,782	C	RUVA	P0A809	Holliday junction branch migration complex subunit RuvA		
ECK1861	*ruvB*		1,939,152	1,940,162	C	RUVB	P0A812	Holliday junction branch migration complex subunit RuvB		
ECK1864	*ruvC*		1,941,661	1,942,182	C	EG10925	P0A814	Crossover junction endodeoxyribonuclease RuvC	nuclease	
ECK2005	*sbcB*	*exoI, cpeA, xonA*	2,076,786	2,078,213		EG10926	P04995	exodeoxyribonuclease I	nuclease	
ECK0391	*sbcC*	*rmuA*	408,612	411,758	C	EG10927	P13458	ATP-dependent structure-specific DNA nuclease—SbcC subunit	nuclease	
ECK0392	*sbcD*	*yajA*	411,755	412,957	C	EG11094	P0AG76	ATP-dependent structure-specific DNA nuclease—SbcD subunit	nuclease	
ECK4051	*ssb*	*exrB, lexC*	4,264,602	4,265,138		EG10976	P0AGE0	ssDNA-binding protein		
ECK3833	*tatD*	*yigX, yigW, mttC*	4,017,463	4,018,245		EG11481	P27859	3’ → 5’ ssDNA/RNA exonuclease TatD (exonuclease XI)	nuclease	
ECK1172	*umuC*	*uvm*	1,227,191	1,228,459		EG11056	P04152	DNA polymerase V catalytic protein	polymerase	
ECK1171	*umuD*		1,226,772	1,227,191		EG11057	P0AG11	DNA polymerase V protein UmuD	polymerase	
ECK3806	*xerC*		3,990,196	3,991,092		EG11069	P0A8P6	Site-specific tyrosine recombinase	recombinase	
ECK2889	*xerD*	*xprB*	3,032,755	3,033,651	C	EG11071	P0A8P8	Site-specific recombinase	recombinase	
ECK2505	*xseA*	*xse*	2,628,140	2,629,510		EG11072	P04994	exodeoxyribonuclease VII subunit XseA	nuclease	
ECK0416	*xseB*	*yajE*	437,106	437,348	C	EG11098	P0A8G9	exodeoxyribonuclease VII subunit XseB	nuclease	
ECK1747	*xthA*	*xth*	1,827,234	1,828,040		EG11073	P09030	exodeoxyribonuclease III	nuclease	
ECK0535	*ybcK*		564,907	566,433		G6300	P77698	Predicted recombinase YbcK	recombinase	DLP12 prophage
ECK2639	*yfjX*		2,769,827	2,770,285		G7378	P52139	Predicted antirestriction protein YfjX	nuclease	CP4-57 prophage
ECK2793	*ygdG*	*xni, exo*	2,924,963	2,925,718		EG12372	P38506	Ssb-binding protein; misidentified as ExoIX; synthetic lethal with polA; no nuclease activity detected; flap endonuclease family protein	nuclease	

ECK: ECK ID defined by Riley et al., PubMed: 16397293; Gene: gene name; Synonym: alternate name; (Left, Right): coordinates on BW25113 genome; Ori: orientation on BW25113 genome (blank: clockwise, C: counterclockwise); EcoCyc: Entry ID PubMed: 30406744, EG IDs PMC3531124; UniProtKB: Entry ID PubMed ID: 18287689; Description: annotation of gene and gene product; Class: protein functional classification; Origin: ancestral origin.

**Table 2 microorganisms-10-01835-t002:** Classification of genome modification methods in *E. coli*.

Recombination Category	System	Factors	Reference
Homologous	*recBCD*	Exonuclease VIII	[32]
	λ Red recombinase	Red recombinase	[30]
Site-specific	*Int-att*	Integrase, *att* site	[33]
	Cre-*lox*	Cre site-specific recombinase	[34]
	FLP-FRT	Flippase, FRT-site	[35]
Mobile element	Mu phage		[36]
	Tn5		[37]
	Tn7		[38]
	Tn10		[39]
	Mariner Tn		[40]
	Group II intron		[41]

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
