# Peer review of "Past, Present, and Future of Genome Modification in Escherichia coli"

_microorganisms, 2022, doi:10.3390/microorganisms10091835_

Round 1
Reviewer 1 Report (Previous Reviewer 2)
This review is devoted to the subject of genomic modification of Escherichia coli. The review is structured systematically correctly and allows you to easily get to the heart of the subject. Significant advantages are the volume of literary sources, graphic material and the systematization of all material in tables. This makes it much easier for readers to find answers to many questions that arise. The relevance of this review is beyond doubt. Such reviews are important and needed, especially in this area of ​​microorganism sciences. The article clearly fits into the subject of this journal. The good sense of humor of the authors should be noted. There are some minor points for improvement:
1. Abstract. Can be expanded.
2. The introduction can be expanded by also pointing out other reviews in this topic and how the review of the authors is better than others.
3. Unify the drawings, please.
4. Epilog can also be modified. Perhaps the authors will point out the prospects (as they represent) of this direction.
Author Response
please check the attachment

Reviewer 2 Report (New Reviewer)
The present review well summarizes the genomic engineering with the E. coli cells. It provides the readers with a historical overview of the developments in genomic technologies and their related applications. I have the following minor comments for the authors to improve the manuscript’s organization.
1) Main text:
L96-98, commercially; please delete “An improved E. coli …”.
L264, blank page?
L312-316, repeated and commercially; please consider deleting or combining it to L93-96.
L320, inappropriate; please consider changing “minimal” to “small” or “reduced”.
L328-329, commercially; please delete “This deletion project …”.
L476, improper; please consider changing “a separate study” to “another study” or “an alternative study”.
L483, improper; please consider changing “As of” to “In”.
L508-514, not informative as a single section; please consider combining section 4 with section3.2.2.
L572-598, largely repeated; please consider combining it with section 3 (maybe section 3.4).
L588-589, confusing; please delete “The saying in Japan…”.
L633, what does “comprehensive resources” mean/represent? Please rephrase or explain it.
L639, improper usage; please delete “however”.
L643, discussing whether an artificial cell is microbial or not is inappropriate; please delete “microbial”.
L649, redundant; please delete “This research was founded by”.
2) Figures:
An additional figure that outlines the achievements of genomic engineering (flowchart regarding sections 2 and 3) is appreciated; as an example, Fig. 1 in microorganisms 8010003.
Figure 3, poor resolution; please replace it.
Figure 4, please reduce the image size to fit the frame.
Author Response
please check the attachment

This manuscript is a resubmission of an earlier submission. The following is a list of the peer review reports and author responses from that submission.
Round 1
Reviewer 1 Report
The review article Past, Present, and Future of genome modification in Escherichia coli summarized the history and evolvement of genome modification research in E. coli. The paper extensively described and compared different homologous recombination technologies that have been applied to modify the genome of E coli. The article also discussed the future direction in the field. This work could potentially be useful for readers in the field of microbiology and particular in bacterial or mycobacterial research. However, unfortunately I found the paper was hard to read due to grammar issues and a lack of logic flow. Especially when the article is long, the logic flow is extremely important for readers so that the theme of the paper can be conveyed. I recommend this paper to go through extensive grammar checks and re-structure in order to engage the readers, and make the paper clearer and easy to follow.
I have found many sentences that need to be checked for grammar or the main idea is vague. I point out some issues of this paper that need to be addressed, and the issues are not limited to what I have listed here.
- The first sentence in the Introduction is not Grammarly correct.
- Page 2, line 57 to 70. This entire paragraph is hard to follow. The logic flow does not seem to be reasonable. The use of “however… furthermore… however…” seems to be very confusing to me.
- The Section 2 is really long and tedious. It includes some repetitive parts that have been mentioned before in the text. I would recommend to shorten it. For example, line 84 to 93 is repeating what have been discussed before in the introduction.
- I recommend the authors to use subtitles within section 2 to summarize the main ideas that you want to convey and to make the text easy to follow.
- Page 3, Line 98-99, check this sentence.
- Page 6, line 11-16, this paragraph seems to lack of context to me.
- Page 6, line 22-23, check this sentence.
- Figure 1 and figure 2 have really low resolutions. It was hard to read the texts in the figures.
- Page 12, line 132-page 13 line 146, is this paragraph the legend for figure 3? Reformat.
- Page 13 line 151-159, is this paragraph the legend for figure 4? Or is it a part of the main text? Reformat.
- Page 14, line 163-176, is this paragraph the legend for figure 5? Or is it a part of the main text? Reformat. It is confusing to me.
- Page 16. Section 6, I recommend the authors to use subtitles within section 6 to summarize the main ideas that you want to convey and to make the text easy to follow.
- Page 13, line 446, what technology is “this technology” referring to?
- Page 13, line 449-450, check this sentence.
- Page 22, the discussion part is really tedious and some paragraphs lack of theme and main ideas. For example, line 576-591, can the author summarize what you would like to convey here? I highly recommend to shorten this entire section.
Reviewer 2 Report
This review is devoted to genomic modifications of Escherichia coli. The article is written in an accessible and understandable language, it is relevant as a systematization of knowledge, in addition, it is replete with illustrative material that is easy to understand. Thanks to this, it is quite easy to navigate this material, which is a significant advantage of this review. In my opinion, there are very minor points that can be improved:
1. Abstract. It is desirable to describe more voluminously. Please complete this.
2. Page 7 is completely blank. This is important to fix.
3. Figure 1 has a very significant caption. Here it is possible to either separate the drawings, or something else.
4. Make Figure 4 more readable. It's too small.